# Assessment of Heavy Metal Pollution Levels in Sediments and of Ecological Risk by Quality Indices, Applying a Case Study: The Lower Danube River, Romania

Valentina Andreea Calmuc [1], Madalina Calmuc [1], Maxim Arseni [1], Catalina Maria Topa [1], Mihaela Timofti [1], Adrian Burada [2], Catalina Iticescu [1,*] and Lucian P. Georgescu [1]

1   European Center of Excellence for the Environment, Faculty of Sciences and Environment, University of Galati, 800001 Galati, Romania; valentina.calmuc@ugal.ro (V.A.C.); madalina.calmuc@ugal.ro (M.C.); maxim.arseni@ugal.ro (M.A.); catalina.topa@ugal.ro (C.M.T.); mihaela.timofti@ugal.ro (M.T.); lucian.georgescu@ugal.ro (L.P.G.)
2   Danube Delta National Institute for Research and Development, 820112 Tulcea, Romania; adi_burada@yahoo.com
*   Correspondence: catalina.iticescu@ugal.ro; Tel.: +40-729-041-801

**Abstract:** It is a well–known fact that heavy metal pollution in sediments causes serious problems not only in the Danube basin, but also in the large and small adjacent river streams. A suitable method for assessing the level of heavy metals and their toxicity in sediments is the calculation of pollution indices. The present research aims to assess heavy metal pollution in the Lower Danube surface sediments collected along the Danube course (between 180 and 60 km) up to the point where the Danube River flows into the Danube Delta Biosphere Reserve (a United Nations Educational, Scientific and Cultural Organization—UNESCO, protected area). In addition, this monitored area is one of the largest European hydrographic basins. Five heavy metals (Cd, Ni, Zn, Pb, Cu) were analyzed in two different seasons, i.e., the autumn of 2018 and the spring of 2019, using the Inductively Coupled Plasma Mass Spectrometry (ICP– MS) technique. Our assessment of heavy metal pollution revealed two correlated aspects: 1. a determination of the potential risks of heavy metals in sediments by calculating the Potential Ecological Risk Index (RI), and 2. an evaluation of the influence of anthropogenic activities on the level of heavy metal contamination in the surface sediments, using three specific pollution indices, namely, the Geo–Accumulation Index (*Igeo*), the Contamination Factor (*CF*), and the Pollution Load Index (*PLI*). The results of this pioneering research activity in the region highlighted the presence of moderate metal (Ni and Cd) pollution and a low potential ecological risk for the aquatic environment.

**Keywords:** lower Danube River; sediment pollution indices; heavy metals; potential ecological risk

## 1. Introduction

The assessment of heavy metal concentration in aquatic ecosystems represents a topic of interest due to heavy metal's toxicity and their special property of bio-accumulation in organisms [1–3]. A total of 23 heavy metals are commonly found in high concentrations in the environment, which can become toxic and dangerous [4], and it is known that population exposure to heavy metals may cause serious medical problems, such as cancer, organ and nervous system damage, autoimmunity, and even death in some instances [5]. The five metals studied in this paper (Cd, Ni, Zn, Pb, Cu) are the most frequently identified heavy metals in the environment, which, in high concentrations, are considered toxic to ecosystems and human health. For example, studies have shown that human exposure to high levels of Cd causes kidney disease, infertility, mental, intestinal disorders and cancer [6]. Nickel intoxication determines respiratory dysfunction, heart disorders and cancer [7]. The ingestion of Zn high levels can cause pancreatic complications, anaemia and

stomach pain [8]. The most common adverse effects of Pb in humans are neuronal dysfunction (especially in children), deficits in renal function, hypertension and heart disease, and disorders of the reproductive system [9]. Exposure to high concentrations of Cu can cause liver, kidney and gastrointestinal disease, damage to the immune system, Wilson's disease and anxiety [10,11]. Moreover, contamination of the aquatic environment with these heavy metals can affect aquatic biodiversity, thus causing ecological imbalances [12].

The Danube is a complex aquatic ecosystem, which hosts a large variety of flora and fauna [13], and is likely to be exposed to heavy metal pollution. Since the long biological half-life of heavy metals in the aquatic environment is a major problem at present, recent studies have shown an interest in reducing pollution sources and the toxic action of heavy metals on different types of aquatic life—more precisely, on ichthyofauna, benthic fauna and macrophytes [14]. In addition, several studies have been carried out, highlighting the importance of periodic monitoring of pollutants in aquatic ecosystems using various analysis techniques—in situ (sensor technology) [15,16] or ex situ (spectrometric, chromatographic methods) [17]—to perform a complex environmental assessment.

The main sources of heavy metal pollution along the Lower Danube originate in municipal waste, sewage discharge, pesticides, fertilizers, the burning of fossil fuels and a series of navigation and mining activities [18–21]. Under these circumstances, monitoring of the physico-chemical and biotic qualities of the Danube water should be doubled by an evaluation of the sediments' quality and the development of specific evaluation measures. This means, in our case, evaluating the degree of heavy metal pollution in the Lower Danube water and identifying the appropriate indices which can be used to determine the quality of surface sediments.

As a natural component of the aquatic ecosystem, the sediment serves as a reservoir for a wide variety of pollutants [22,23]. Therefore, the excessive presence of heavy metals loads in the sediment from anthropogenic impact can pose a threat to the water supply and produce changes in environmental conditions. This aspect must be taken into account, as the Danube River is the primary water supply for three major cities in south–east Romania, namely, Braila, Galati and Tulcea [24]. In addition, heavy metal contamination in sediments has significant implications for benthic organisms, biota and water quality and numerous invertebrates who process sediments as a food source. Since heavy metals can be bioaccumulated in invertebrate organisms, such metals may subsequently reach the other components of the trophic chain [25,26].

The main aims of this research article are to evaluate and quantify the influence of anthropic activities on the level of heavy metal contamination in the surface sediments of the Lower Danube, and to assess the potential risks to which the aquatic ecosystem is exposed. In order to reach these aims, different specific pollution indices, which are important in assessing the quality of large-stream water bodies, were used and tested, namely the geo–accumulation index (*Igeo*), contamination factor (*CF*), Pollution Load Index (*PLI*) and the Potential Ecological Risk Index (*RI*). The scientific originality of this paper is represented by the study area, as these quality indices have not been calculated, although there are a wide variety of heavy metal pollution sources nearby. In addition, the investigation area is directly connected to the Danube Delta Biosphere Reserve, which is characterized by rich, UNESCO-protected flora and fauna. For this reason, it is very important to periodically monitor the quality of sediments in this sector.

## 2. Materials and Methods

### 2.1. Study Area

A total of 15 sampling stations were selected to assess the level of heavy metals sediment contamination (Figure 1), according to the existing pollution sources located along the lower Danube River, between 180 and 60 km. In this area, the Danube River crosses three major cities in Romania (Brăila, Galati and Tulcea), with a large number of inhabitants and significant industrial activity (Damen Galati, Navrom Galati, Vard Braila, Vard Tulcea). The monitoring and evaluation of surface sediments quality is important

because, in this perimeter, the Danube flows into the Danube delta and feeds the largest variety of lakes and canals, hosting a huge variety of fauna and flora unique to Europe.

| ID sampling point | Name | Longitude (deg°min'sec") | Latitude (deg°min'sec") |
|---|---|---|---|
| S4 | Chiscani | 45°10'32.18"N | 27°56'33.06"E |
| S3 | Bac 2 (Bra | 45°14'24.44"N | 27°57'54.32"E |
| S2 | Bac 1 (Bra | 45°17'47.73"N | 27°59'47.63"E |
| S1 | Insula Chic | 45°20'33.70"N | 28°1'1.69"E |
| S5 | Priza Duna | 45°23'18.92"N | 28°1'26.57"E |
| S6 | Siret | 45°24'10.59"N | 28°1'52.09"E |
| S7 | Libertatea | 45°25'42.74"N | 28°3'41.75"E |
| S8 | Cotul Pisic | 45°24'56.66"N | 28°11'32.58"E |
| S9 | Prut | 45°27'58.44"N | 28°12'43.63"E |
| S10 | Grindu | 45°27'0.80"N | 28°16'17.02"E |
| S11 | Luncavita | 45°19'53.15"N | 28°20'3.27"E |
| S12 | Isaccea | 45°16'29.26"N | 28°29'31.10"E |
| S13 | Somova | 45°14'11.64"N | 28°39'50.06"E |
| S14 | Vard Amor | 45°12'41.64"N | 28°47'27.15"E |
| S15 | Tulcea | 45°10'54.96"N | 28°48'9.62"E |

**Figure 1.** Sampling Stations along the lower Danube River.

Sediment samples were collected monthly over two different seasons, i.e., the autumn of 2018 and the spring of 2019. Considering that there were no significant variations in the heavy metal concentrations in the surface sediments during the three months of a season, the seasonal average was taken into account in the calculation of the pollution indices.

*2.2. Sample Collection and Analysis*

The surface sediment samples were collected using a Van Veen Grab Sampler (KC Denmark A/S, Silkeborg, Denmark) from the first 10 cm of the sediment surface and deposited in polyethylene recipients. During the transport and temporary storage (1/2 days), the sediment samples were preserved at 4 °C. In the preliminary stage, sediment samples were dried at 105 °C until they reached a constant weight and were sieved using a 125 mm sieve.

Sediment samples were transported and analyzed in the Chemistry Laboratory of the Danube Delta National Institute for Research and Development, Tulcea, Romania. The mineralization of the samples was performed by the Anton Paar microwave digestion system. After the sample preparation stage, the heavy metal concentrations (Pb, Cu, Cd, Zn, Ni) were determined in accordance with the standard SR EN ISO 17294–2, 2005. The analyses were performed using Perkin Elmer ICP–MS Elan DRC–e (PerkinElmer LAS (UK)Ltd, Seer Green, England, UK) [1]. Sediment samples were analyzed in five replicates, for which the relative standard deviations (%RSDs) were less than 10% of the trace elements. The calibration curve was made of six points and the calibration standard solutions were

prepared by successive dilution of a high-purity ICP-multi-element calibration standard (10 µg/mL, batch N9301720, Matrix: 5% HNO3, PerkinElmer). The methods and results were validated using a Sigma–Aldrich Certified Reference Materials–Metals in soil—batch LRAC3749, PRODUCT ID SQC001. The accuracy of the performed analyses was tested using CRM Metals in soil, batch LRAC3749, PRODUCT ID SQC001 (Merck Romania SRL, Bucharest, Romania, an affiliate of Merck KGaA, Darmstadt, Germany). The percentage ranged between 97 and 114% for Pb, from 86 to 104% for Cu, from 87 to 110% for Cd, from 96 to 114% for Zn, and from 95 to 108% for Ni. Table 1 specifies the $R^2$ values obtained for each heavy metal analyzed.

**Table 1.** $R^2$ values of each heavy metal.

| Element | $R^2$ |
|---------|-------|
| Pb | 0.9997 |
| Cu | 0.9998 |
| Cd | 0.9997 |
| Zn | 0.9999 |
| Ni | 0.9998 |

*2.3. Methods for Assessing Anthropogenic Contributions to Heavy Metal Sediment Pollution*

2.3.1. Calculation of the Geo–Accumulation Index (*Igeo*)

The Geo–Accumulation Index was proposed by Müller (1969) [27] to assess the pollution levels of each heavy metal in surface sediments, taking their background value into account [28,29]. According to Litenithy and Laszlo (1999), Woitke et al. (2003), and Ilie et al. (2017) [30–32], the background values of heavy metals in sediments from the Danube are: 35, 0.25, 25, 10, and 130 mg kg$^{-1}$ for Cu, Cd, Pb, Ni and Zn.

The *Igeo* was determined using the following equation (Equation (1)) [33–35]:

$$Igeo = \log2 \frac{Cn}{K * Bn} \tag{1}$$

where *Igeo* is the index of geo accumulation for each heavy metal; *Cn* is the concentration of heavy metals determined in the sediment sample; *Bn* refers to the background value of heavy metals, *K* = 1.5 represents a constant, which compensates for weathering and lithogenic effects [36].

The values of the *Igeo* allow for an evaluation of the pollution intensity with heavy metals, according to Table 2.

**Table 2.** Level of pollution with heavy metals according to *Igeo* [37,38].

| Igeo Value | Class | Pollution Intensity |
|------------|-------|---------------------|
| *Igeo* ≤ 0 | 0 | unpolluted |
| 0 < *Igeo* < 1 | 1 | unpolluted to moderately polluted |
| 1 < *Igeo* < 2 | 2 | moderately polluted |
| 2 < *Igeo* < 3 | 3 | moderately to strongly polluted |
| 3 < *Igeo* < 4 | 4 | strongly polluted |
| 4 < *Igeo* < 5 | 5 | strongly to very strongly polluted |
| *Igeo* ≥ 5 | 6 | very strongly polluted |

2.3.2. Calculation of the Contamination Factor (*CF*)

The contamination factor describes the pollution level of sediment with a given heavy metal and is calculated as the ratio between the concentration of each measured heavy metal (*Cn*) and its background value (*C$_{bn}$*) (Equation (2)) [39,40]

$$CF = \frac{Cn}{C_{bn}} \tag{2}$$

Based on the results obtained for *CF*, the level of heavy metal contamination is established according to Table 3.

**Table 3.** Contamination level of sediment according to the *CF* value [41].

| *CF* Value | Contamination |
|:---:|:---:|
| $CF < 1$ | Low |
| $1 \leq CF < 3$ | Moderate |
| $3 \leq CF < 6$ | Considerable |
| $CF > 6$ | Very high |

### 2.3.3. Calculation of the Pollution Load Index (*PLI*)

The *PLI* is a tool used to assess the global level of sediment contamination, taking the concentrations of several heavy metals into account. This is calculated based on the *CF* of each metal (Equation (3)) [42,43]

$$PLI = \left( CF_{Me1} \times CF_{Me2} \times \ldots \times CF_{Men} \right)^{1/n} \tag{3}$$

where *PLI* is the pollution load index, $CF_{Me1,2,3,\ldots,n}$ represents the contamination factor of each metal *Me*1, 2, 3, . . . , *n* and *n* is the number of metals.

The values of *PLI* < 1 indicate the absence of heavy metal contamination, whereas *PLI* > 1 shows the presence of heavy metal pollution [32,44].

### 2.4. Method for Assessing the Potential Risks of Heavy Metals
Calculation of Potential Ecological Risk Index (*RI*)

The *RI* was calculated to assess the potential risks from heavy metals from surface sediments. This index was developed by Hakänson (1980) [45] to evaluate the potential risk of heavy metal contamination in sediments. This method takes the toxicity and combined effects of heavy metals on the aquatic ecosystem into account [46]. According to Hakänson (1980) the toxic response factors for the analyzed heavy metals, such as Pb, Cu, Cd, Zn and Ni, are 5, 5, 30, 1 and 5. The final value of *RI* is obtained by calculating the following formulas [47–53]

$$RI = \sum Er^{Me} \tag{4}$$

$$Er^{Me} = Tr^{Me} \times CF^{Me} \tag{5}$$

$$CF^{Me} = \frac{C^{Me}}{C_{SCM}^{Me}} \tag{6}$$

where *RI* (Table 4 lists the levels of ecological risk according to the obtained *RI* index value) is the sum of potential risk of individual heavy metal; $Er^{Me}$ is the potential ecological risk of individual metal Me; $Tr^{Me}$ refers to the toxic-response factor for each metal Me; $CF^{Me}$ is the contamination factor for each metal Me; $C^{Me}$ is the measured level of heavy metal in the sediment; $C_{SCM}{}^{Me}$ is the standard value of each heavy metal concentration according to the Romanian Order 161/2006 (Table 5).

**Table 4.** Values of the *RI* [54].

| $Er^{Me}$ Value | RI Value | Ecological Risk Level |
|:---:|:---:|:---:|
| $Er^{Me} \leq 40$ | $RI < 150$ | Low |
| $40 < Er^{Me} \leq 80$ | $150 \leq RI < 300$ | Moderate |
| $80 < Er^{Me} \leq 160$ | $300 \leq RI < 600$ | Considerable |
| $160 < Er^{Me} \leq 320$ | | High |
| $Er^{Me} > 320$ | $RI \geq 600$ | Very high |

**Table 5.** Standard value of each heavy metal concentration in the sediment according to the Romanian Order 161/2006 [47].

| Heavy Metal | Standard Value (mg·kg$^{-1}$) |
| --- | --- |
| Cd | 0.8 |
| Cu | 40 |
| Pb | 85 |
| Zn | 150 |
| Ni | 35 |

Table 6 shows the average values of heavy metal concentrations and standard deviations, obtained during the two seasons studied (autumn of 2018 and spring of 2019), in the 15 monitoring stations. Based on these results, the indices presented above were calculated. The obtained results indicate that the concentrations of the five metals measured in the Danube surface sediments are increasing in the sequence of Cd < Pb < Cu < Ni < Zn.

In order to assess the spatial distribution of the heavy metal contamination level in the surface sediments, the values obtained for the *Igeo*, the *PLI* and the *RI* were represented in the form of pollution maps (Figures 2, 3 and 5).

**Table 6.** Seasonal average of heavy metal concentrations and standard deviation.

| | Sites | | | S1 | S2 | S3 | S4 | S5 | S6 | S7 | S8 | S9 | S10 | S11 | S12 | S13 | S14 | S15 |
|---|---|---|---|---|---|---|---|---|---|---|---|---|---|---|---|---|---|---|
| Heavy metal concentrations (mg·kg⁻¹) | Pb | Autumn | Mean | 5.90 | 8.96 | 12.57 | 8.49 | 7.93 | 5.17 | 7.28 | 21.14 | 4.84 | 7.55 | 7.83 | 6.84 | 8.29 | 8.11 | 5.34 |
| | | | Std.dev | 0.15 | 0.64 | 0.82 | 0.56 | 0.26 | 0.21 | 0.45 | 1.60 | 0.48 | 0.12 | 0.42 | 0.52 | 0.57 | 0.63 | 0.38 |
| | | Spring | Mean | 6.02 | 6.05 | 13.78 | 6.41 | 5.70 | 4.17 | 5.68 | 10.35 | 4.31 | 8.33 | 14.64 | 9.87 | 9.90 | 6.76 | 8.01 |
| | | | Std.dev | 0.44 | 0.23 | 0.55 | 0.18 | 0.43 | 0.28 | 0.32 | 0.63 | 0.27 | 0.50 | 0.22 | 0.39 | 0.61 | 0.59 | 0.52 |
| | Cu | Autumn | Mean | 4.30 | 10.72 | 17.39 | 12.58 | 12.54 | 7.60 | 11.81 | 10.24 | 7.89 | 13.42 | 6.68 | 9.79 | 16.64 | 15.17 | 9.42 |
| | | | Std.dev | 0.11 | 0.87 | 0.72 | 0.39 | 0.68 | 0.46 | 0.34 | 0.49 | 0.25 | 1.06 | 0.38 | 0.44 | 0.53 | 0.49 | 0.27 |
| | | Spring | Mean | 10.31 | 11.65 | 25.01 | 8.97 | 10.08 | 7.55 | 7.47 | 19.47 | 9.29 | 17.18 | 27.50 | 20.75 | 23.29 | 9.29 | 10.07 |
| | | | Std.dev | 0.80 | 0.41 | 0.49 | 0.40 | 0.64 | 0.44 | 0.72 | 0.28 | 0.33 | 1.11 | 0.88 | 0.39 | 0.68 | 0.48 | 0.66 |
| | Cd | Autumn | Mean | 0.30 | 0.59 | 0.74 | 0.50 | 0.50 | 0.46 | 0.57 | 0.54 | 0.54 | 0.76 | 0.53 | 0.63 | 0.75 | 0.57 | 0.53 |
| | | | Std.dev | 0.006 | 0.025 | 0.053 | 0.016 | 0.044 | 0.012 | 0.028 | 0.036 | 0.015 | 0.023 | 0.044 | 0.028 | 0.034 | 0.016 | 0.019 |
| | | Spring | Mean | 0.59 | 0.65 | 0.99 | 0.63 | 0.57 | 0.41 | 0.46 | 0.78 | 0.46 | 0.65 | 0.72 | 0.77 | 0.82 | 0.48 | 0.52 |
| | | | Std.dev | 0.027 | 0.015 | 0.032 | 0.04 | 0.043 | 0.036 | 0.019 | 0.029 | 0.035 | 0.041 | 0.043 | 0.036 | 0.033 | 0.027 | 0.038 |
| | Zn | Autumn | Mean | 58.84 | 118.54 | 120.76 | 87.43 | 84.15 | 62.39 | 84.65 | 77.64 | 64.48 | 146.23 | 85.40 | 96.11 | 121.38 | 117.01 | 69.97 |
| | | | Std.dev | 1.34 | 1.27 | 1.32 | 0.98 | 1.46 | 1.55 | 1.68 | 1.32 | 1.08 | 1.56 | 0.98 | 1.44 | 1.39 | 1.70 | 1.03 |
| | | Spring | Mean | 78.69 | 84.21 | 177.33 | 73.57 | 95.67 | 71.27 | 63.21 | 131.50 | 66.06 | 121.05 | 161.24 | 146.53 | 154.34 | 81.26 | 86.43 |
| | | | Std.dev | 0.78 | 1.22 | 0.96 | 0.87 | 1.33 | 1.62 | 1.39 | 0.85 | 0.52 | 1.31 | 0.99 | 1.63 | 1.28 | 1.75 | 1.08 |
| | Ni | Autumn | Mean | 16.03 | 29.12 | 27.88 | 22.31 | 20.53 | 14.00 | 19.99 | 22.09 | 16.28 | 23.90 | 24.83 | 24.76 | 28.35 | 38.81 | 16.94 |
| | | | Std.dev | 0.96 | 1.69 | 0.98 | 1.77 | 1.32 | 1.24 | 0.96 | 1.08 | 1.37 | 1.54 | 1.80 | 1.30 | 1.96 | 1.33 | 1.04 |
| | | Spring | Mean | 20.17 | 19.33 | 35.80 | 16.04 | 24.58 | 19.09 | 17.65 | 28.49 | 17.40 | 27.41 | 50.46 | 32.55 | 39.03 | 25.85 | 28.13 |
| | | | Std.dev | 0.85 | 0.98 | 1.32 | 0.96 | 0.58 | 1.36 | 1.26 | 1.44 | 1.28 | 1.63 | 1.82 | 0.96 | 1.22 | 1.14 | 0.85 |

## 3. Results and Discussion

### 3.1. Spatial Distribution of Geo–Accumulation Index (Igeo)

Based on the *Igeo* values obtained in autumn 2018, a moderate to strong cadmium pollution can be observed in stations: S3, S10 and S13 (Figure 2). Usually, industry, and the combustion of fossil fuels and agriculture (phosphoric fertilizers) represent the main sources of environmental pollution with cadmium [4,55–57]. During spring, the highest values of *Igeo* Cd (2 < *Igeo* <3) were recorded in stations S3, S8, S11, S12, S13. These monitoring stations are located near the ferry crossing (S3) and the agricultural lands (S8, S11, S12, S13).

The *Igeo* values for Ni ranged between 1.07 (S6) and 2.54 (S14) during the autumn of 2018 and 1.27 (S4)–2.92 (S11) during spring, 2019. According to *Igeo*, the pollution of Ni varies from Class 2 (moderately polluted) to Class 3 (moderately to strongly polluted). For this metal, limit allowed according by the Romanian Order 161/2006 (35 mg kg$^{-1}$) was exceeded in stations S3, S11, S13, S14 (Table 5). Similar results regarding the level of pollution with Ni were reported in the research of Ilie et al. (2017) along the Danube [32]. Across the monitored Lower Danube sector, the presence and persistence of nickel pollution can have several causes, such as transport, industry, municipal and industrial waste [58].

The *Igeo* values of Pb (−1.78–0.34 during autumn and −2.00–1.03 during spring) indicate the level of unpolluted sediment in most of the sampling stations, except for station S8, where the sediment was classified as "unpolluted to moderately polluted" in the autumn of 2018. Additionally, the result of the *Igeo* calculation for autumn season shows that Cu values for all sites were within the uncontaminated class (*Igeo* ≤ 0). On the other hand, two of the 15 stations (S3,S11) were unpolluted to moderately polluted with Cu in the spring season. Regarding the Zn metal, the *Igeo* values ranged from −0.56 (S1) to 0.75 (S10) (during the autumn of 2018) and −0.46 (S7)–1.03 (S3) (during the spring of 2019). The highest value of this index was obtained for station S3 (177.33 mg kg$^{-1}$) where an exceedance of the allowed limit for Zn (150 mg kg$^{-1}$) was registered. The majority of results obtained for this metal indicate the unpolluted to moderately polluted status of the sediment in both monitored periods.

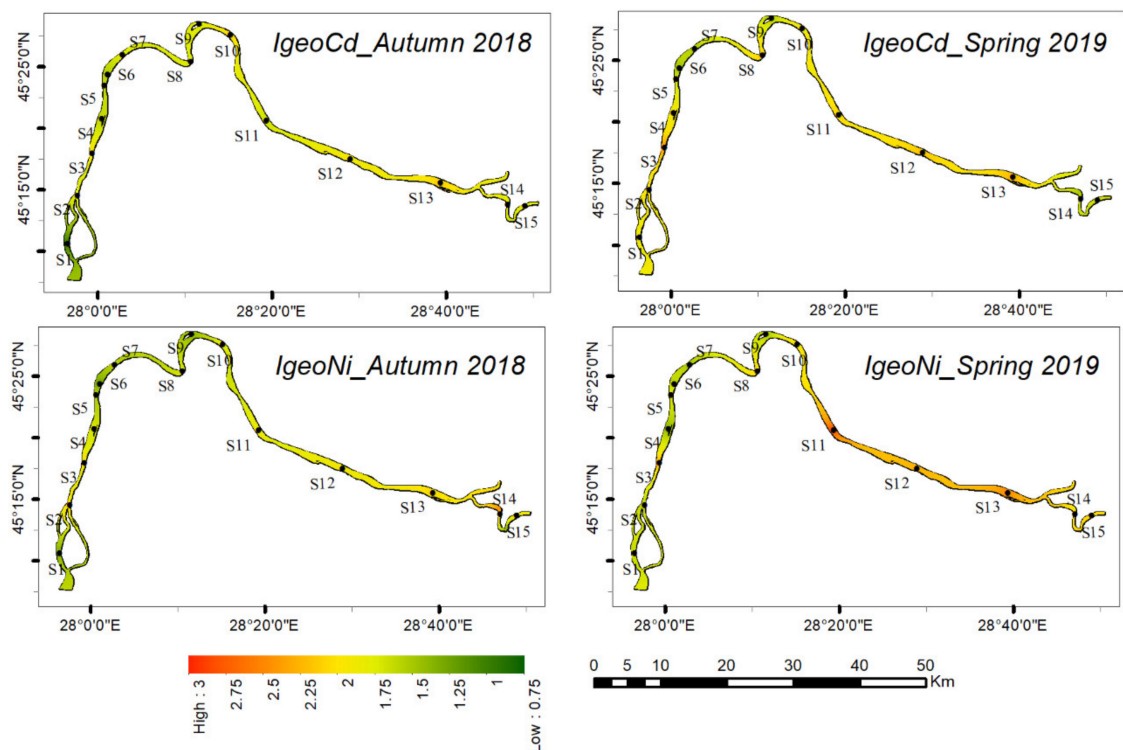

**Figure 2.** *Cont.*

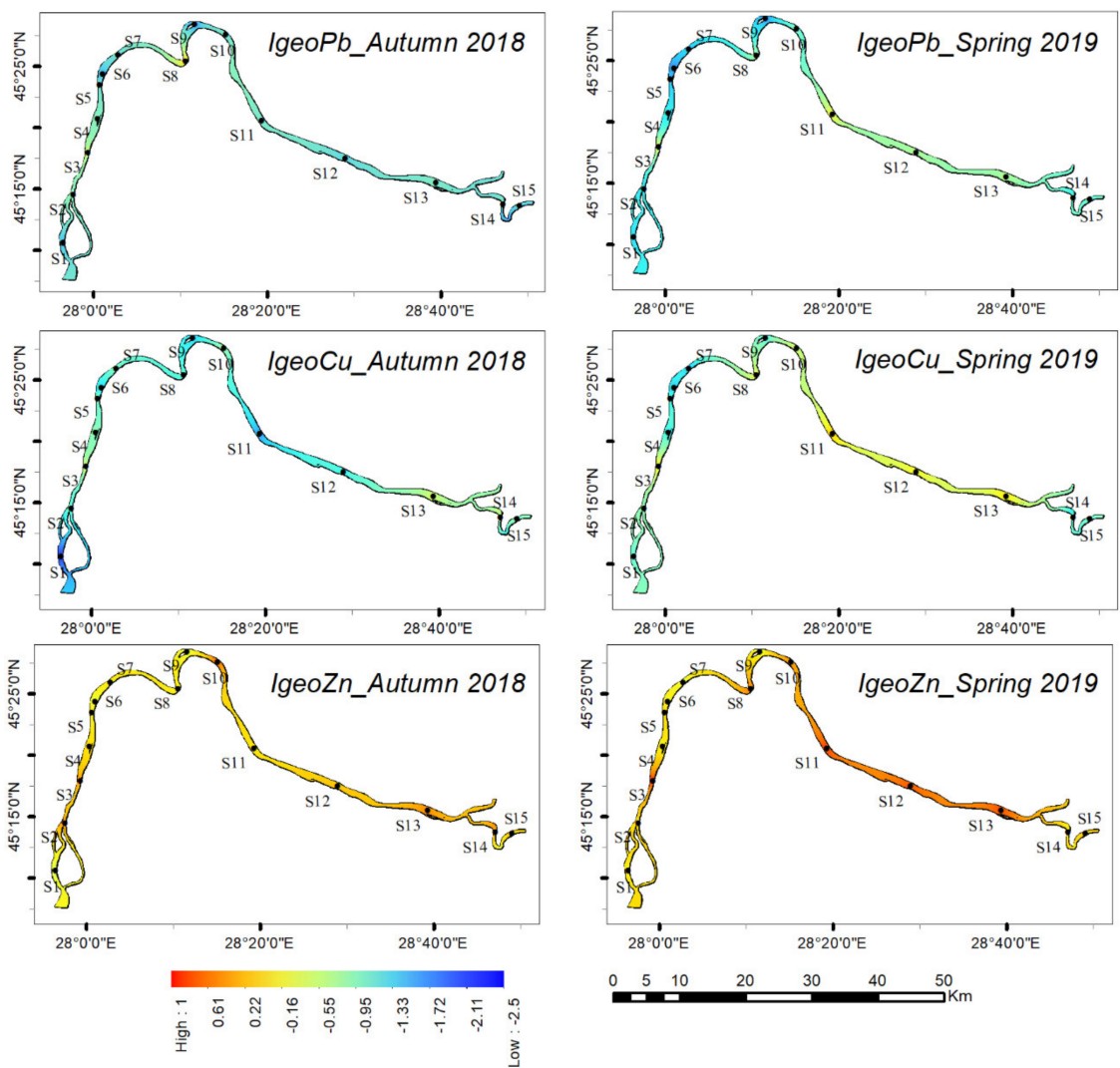

**Figure 2.** Spatial distribution of the *Igeo* for Cd, Ni, Pb, Cu and Zn in the autumn of 2018 and the spring of 2019.

### 3.2. Contamination Factor (CF) and Pollution Load Index (PLI)

The *CF* calculation indicates results similar to the *Igeo* values, but the difference between these two indices is that *Igeo* is used to reflect the degree of sediment contamination of each metal, taking the lithogenic effects, natural fluctuations in metals and some small anthropogenic influences into account [53]. On the other hand, *CF* is a precursor to calculating the *PLI* index, representing an integral part of the formula of this index. In our analysis, the results of the *CF* of each metal show a low contamination of sediments with heavy metals such as Pb and Cu in all the sampling sites, in both monitored seasons ($CF < 1$). For Zn metal, *CF* values (Table 7) indicate low contamination in most monitored stations, with the exception of stations S3, S8, S11–S13, which show moderate contamination ($1 \leq CF < 3$) during the spring season. In agreement with the *CF* index, during the autumn season, all 15 sites were moderately contaminated with Cd, while, during the spring season, only 11 stations reported moderate contamination; the rest (S3, S8, S12, S13) recorded considerable contamination. The highest value of the *CF* was recorded for Ni during the spring season, in monitoring station S11 (5.05). The level of sediment pollution with Ni was moderate in most sites, with the exception of stations S14 (autumn of 2018), S3, and S11–S13 (spring of 2019), where considerable contamination was measured.

**Table 7.** *CF* values.

| Sites | Pb | | Cu | | Cd | | Zn | | Ni | |
|---|---|---|---|---|---|---|---|---|---|---|
| | Autumn | Spring | Autumn | Spring | Autumn | Spring | Autumn | Spring | Autumn | Spring |
| S1 | 0.24 | 0.24 | 0.12 | 0.29 | 1.22 | 2.36 | 0.45 | 0.61 | 1.60 | 2.02 |
| S2 | 0.36 | 0.24 | 0.31 | 0.33 | 2.37 | 2.58 | 0.91 | 0.65 | 2.91 | 1.93 |
| S3 | 0.50 | 0.55 | 0.50 | 0.71 | 2.96 | 3.96 | 0.93 | 1.36 | 2.79 | 3.58 |
| S4 | 0.34 | 0.26 | 0.36 | 0.26 | 1.99 | 2.50 | 0.67 | 0.57 | 2.23 | 1.60 |
| S5 | 0.32 | 0.23 | 0.36 | 0.29 | 2.00 | 2.26 | 0.65 | 0.74 | 2.05 | 2.46 |
| S6 | 0.21 | 0.17 | 0.22 | 0.22 | 1.84 | 1.65 | 0.48 | 0.55 | 1.40 | 1.91 |
| S7 | 0.29 | 0.23 | 0.34 | 0.21 | 2.29 | 1.83 | 0.65 | 0.49 | 2.00 | 1.77 |
| S8 | 0.85 | 0.41 | 0.29 | 0.56 | 2.18 | 3.12 | 0.60 | 1.01 | 2.21 | 2.85 |
| S9 | 0.19 | 0.17 | 0.23 | 0.27 | 2.15 | 1.83 | 0.50 | 0.51 | 1.63 | 1.74 |
| S10 | 0.30 | 0.33 | 0.38 | 0.49 | 3.02 | 2.59 | 1.12 | 0.93 | 2.39 | 2.74 |
| S11 | 0.31 | 0.59 | 0.19 | 0.79 | 2.11 | 2.87 | 0.66 | 1.24 | 2.48 | 5.05 |
| S12 | 0.27 | 0.39 | 0.28 | 0.59 | 2.52 | 3.09 | 0.74 | 1.13 | 2.48 | 3.26 |
| S13 | 0.33 | 0.40 | 0.48 | 0.67 | 2.99 | 3.29 | 0.93 | 1.19 | 2.84 | 3.90 |
| S14 | 0.32 | 0.27 | 0.43 | 0.27 | 2.30 | 1.90 | 0.90 | 0.63 | 3.88 | 2.59 |
| S15 | 0.21 | 0.32 | 0.27 | 0.29 | 2.13 | 2.08 | 0.54 | 0.66 | 1.69 | 2.81 |

Analyzing the spatial distribution of the *PLI* related to the two monitored seasons (the autumn of 2018 and the spring of 2019), it can be observed that the *PLI* values ranged from 0.53 (S1) to 1.17 (S3) in the autumn months, which indicates the absence of heavy metal pollution (*PLI* < 1) in 73% of the monitored stations. During spring 2019, Figure 3 illustrates that *PLI* values ranged from 0.57 (S6) to 1.53 (S11), indicating the presence of sediments heavy metal pollution, especially in monitoring stations S11–S13 (*PLI* > 1).

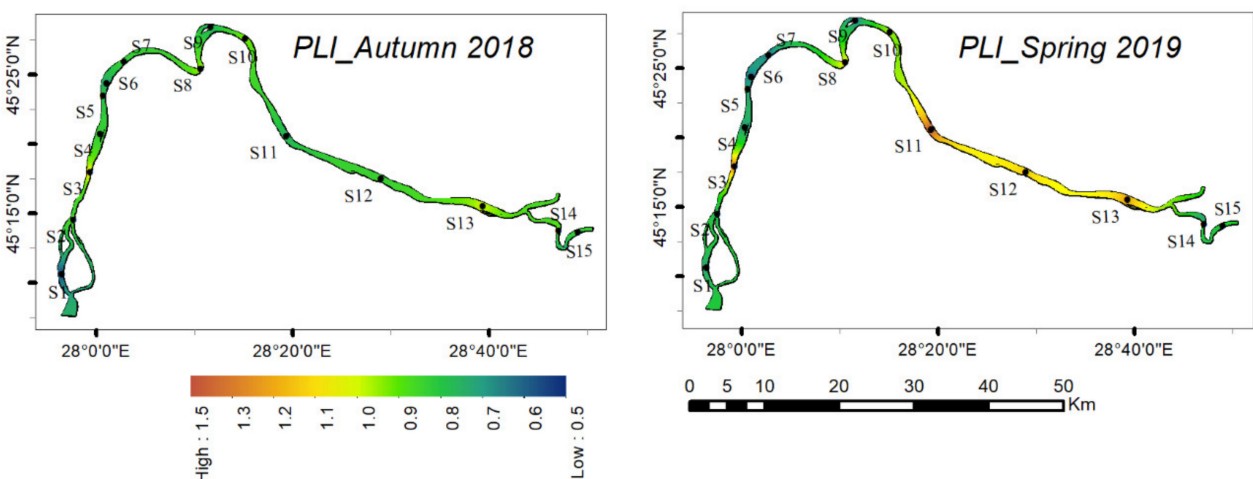

**Figure 3.** Spatial distribution of the *PLI* in the autumn of 2018 and the spring of 2019.

The present study highlights the differences in heavy metal concentrations measured in surface sediments between the monitoring stations and the two seasons. In order to highlight the difference between the two seasons, Principal Component Analysis (PCA) was applied (Figure 4). Figure 4 displays a significantly different distribution between the two seasons of the obtained values, especially for the metals Cd, Cu, Ni and Zn. In contrast, there were no important seasonal variations in Pb concentrations. Moreover, from the PCA statistical analysis, it was observed that, for the heavy metals Cu, Ni, Zn and Cd, higher concentrations were registered during the spring season for most sediment samples.

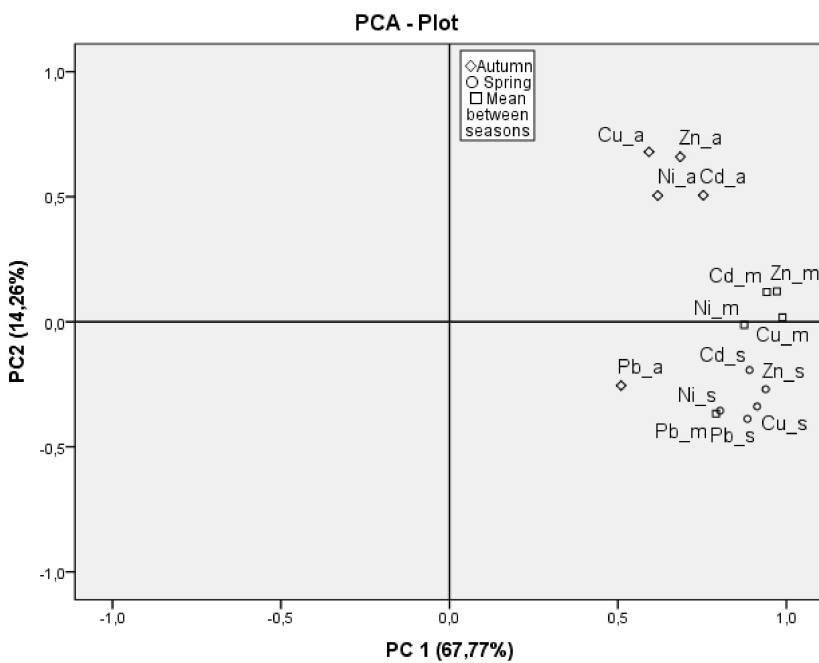

**Figure 4.** The PCA plot of seasonal variations in heavy metal concentrations.

Following the analysis of the seasonal average flows (Table 8), it can be observed that the pollution increases when the average discharge flows are maximum, i.e., in the spring season, when the average flow at the four hydrometric stations reached 5796 m$^3$ s$^{-1}$. With the increase in the discharge, the water velocity also increases; thus, the sediment transport becomes more active. The highwater velocities and high discharge cause the sediments to be transported from upstream to downstream and deposited in the areas where the Danube section has lower flow slopes. Therefore, high concentration values can be observed in sampling points S11–S13, where the flow slope decreases significantly.

**Table 8.** Danube River mean discharge flows (Q$_m$) during autumn of 2018 and spring of 2019.

| Hydrometric Stations | Q$_m$ (m$^3$ s$^{-1}$) | |
| --- | --- | --- |
| | **Autumn of 2018** | **Spring of 2019** |
| Brăila (km 170) | 3220 | 6263 |
| Galați (km 150) | 3677 | 6803 |
| Isaccea (km 103) | 3127 | 6673 |
| Tulcea (km 71) | 1627 | 3443 |
| Average flow | 2913 | 5796 |

Another explanation can be given by the fact that, in the area of points S1–S10, the river sector has a very sinuous path. Therefore, the sedimentation takes place much further downstream, i.e., in the area of points S11–S13, where the course becomes linear, and the width of the riverbed increases. With the increase in the river width, there is a decrease in the flow velocities as well as the depths specific to this area. This favors the deposition of sediments in these places, especially those with high concentrations of heavy metals [59,60].

*3.3. Spatial Distribution of Potential Ecological Risk Index (RI)*

The *RI* index was calculated based on the five heavy metals (Pb, Zn, Cd, Cu and Ni), and the results comprehensively reflect a low ecological risk level for each single element ($Er^{Me} \leq 40$) (Table 9), as well as a low degree of general ecological risk for both monitored seasons (Figure 5). In addition, *RI* results in the surface sediment ranged from 15.00 (S1) to 35.50 (S13) during the autumn season and 17.61 (S6) to 45.96 (S3) during the spring season.

**Table 9.** Potential ecological risk of individual metal ($Er^{Me}$) values.

| Sites | Pb | | Cu | | Cd | | Zn | | Ni | |
|---|---|---|---|---|---|---|---|---|---|---|
| | Autumn | Spring | Autumn | Spring | Autumn | Spring | Autumn | Spring | Autumn | Spring |
| S1 | 0.35 | 0.35 | 0.54 | 1.29 | 11.43 | 22.11 | 0.39 | 0.52 | 2.29 | 2.88 |
| S2 | 0.53 | 0.36 | 1.34 | 1.46 | 22.19 | 24.21 | 0.79 | 0.56 | 4.16 | 2.76 |
| S3 | 0.74 | 0.81 | 2.17 | 3.13 | 27.75 | 37.15 | 0.81 | 1.18 | 3.98 | 5.11 |
| S4 | 0.50 | 0.38 | 1.57 | 1.12 | 18.65 | 23.44 | 0.58 | 0.49 | 3.19 | 2.29 |
| S5 | 0.47 | 0.34 | 1.57 | 1.26 | 18.76 | 21.20 | 0.56 | 0.64 | 2.93 | 3.51 |
| S6 | 0.30 | 0.25 | 0.95 | 0.94 | 17.28 | 15.49 | 0.42 | 0.48 | 2.00 | 2.73 |
| S7 | 0.43 | 0.33 | 1.48 | 0.93 | 21.49 | 17.13 | 0.56 | 0.42 | 2.86 | 2.52 |
| S8 | 1.24 | 0.61 | 1.28 | 2.43 | 20.42 | 29.23 | 0.52 | 0.88 | 3.16 | 4.07 |
| S9 | 0.28 | 0.25 | 0.99 | 1.16 | 20.14 | 17.19 | 0.43 | 0.44 | 2.33 | 2.49 |
| S10 | 0.44 | 0.49 | 1.68 | 2.15 | 28.33 | 24.24 | 0.97 | 0.81 | 3.41 | 3.92 |
| S11 | 0.46 | 0.86 | 0.84 | 3.44 | 19.80 | 26.91 | 0.57 | 1.07 | 3.55 | 7.21 |
| S12 | 0.40 | 0.58 | 1.22 | 2.59 | 23.62 | 28.92 | 0.64 | 0.98 | 3.54 | 4.65 |
| S13 | 0.49 | 0.58 | 2.08 | 2.91 | 28.07 | 30.86 | 0.81 | 1.03 | 4.05 | 5.58 |
| S14 | 0.48 | 0.40 | 1.90 | 1.16 | 21.55 | 17.84 | 0.78 | 0.54 | 5.54 | 3.69 |
| S15 | 0.31 | 0.47 | 1.18 | 1.26 | 19.99 | 19.45 | 0.47 | 0.58 | 2.42 | 4.02 |

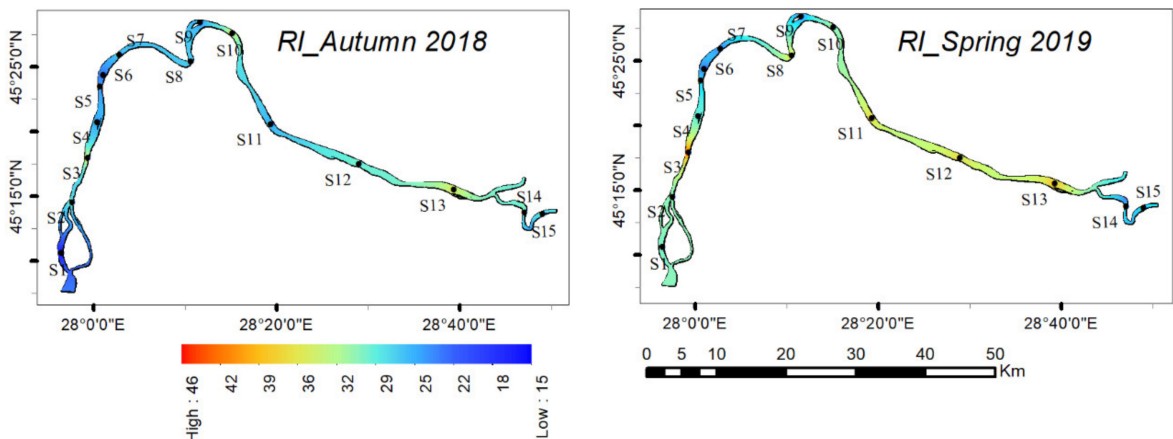

**Figure 5.** Spatial distribution of the *RI* in the autumn of 2018 and the spring of 2019.

The $Er^{Me}$ results of heavy metals in the surface sediments of the Lower Danube are shown in Table 9. The most significant values of *Er* were recorded for Cd, because, according to Hakanson's approach, the toxic response of this metal is the highest. However, it does not represent a high ecological risk in the Lower Danube sediments, due to the fact that the measured Cd values are situated below the permitted limit according to the Romanian Order 161/2006. The second metal that made an important contribution to the final result of the RI index is Ni. Similar to the *PLI* index, the highest *RI* values were recorded during the spring season for stations S3 and S11–S13 (Figure 5).

## 4. Conclusions

In this study, the indices *Igeo*, *CF, PLI* and *RI* were calculated to determine the degree of heavy metal pollution of lower Danube surface sediments and their potential ecological risk.

The results of the *Igeo*, *CF* and the *PLI* indicated that a heavy metal load with Ni and Cd in surface sediments is due to anthropogenic activities, and had an influence on the pollution levels of the lower Danube surface sediments analysed. Generally, *Igeo* and *CF* suggested that the average concentrations of heavy metals analyzed in the sediment were

higher than the background value. However, in most sites, no significant pollution was identified for the heavy metals Pb and Cu.

According to the potential ecological risk of individual metal ($Er^{Me}$), the values of the *RI* index were influenced by the heavy metals in the following sequence: Pb < Zn < Cu < Ni < Cd. However, the *RI* values revealed the existence of a low ecological risk for the surface sediments, as the limits for most heavy metals were not significantly exceeded.

In addition, the results of the pollution indices tested in the present study indicate the existence of temporal and spatial fluctuations regarding the pollution level with certain heavy metals, due to the presence of pollution sources, heavy metal mobility, sediment characteristics, sediment transport, the hydro-morphological profile of the river and climatic conditions [61].

In the future, these indices may be included together with other water quality indices (i.e., WQI) in a global index, to perform a complex assessment of the Danube River quality.

**Author Contributions:** Conceptualization, V.A.C. and C.I.; Data curation, M.C. and M.A.; Formal analysis, M.C., M.T. and M.A.; Funding acquisition, C.I. and L.P.G.; Investigation, V.A.C., C.M.T. and L.P.G.; Methodology, V.A.C., A.B., C.I. and L.P.G.; Writing—original draft preparation, V.A.C., C.M.T. and C.I. All authors have read and agreed to the published version of the manuscript.

**Funding:** This research was funded by the Romanian Ministry of Research and Innovation, by the project DANS, 4/2018.

**Institutional Review Board Statement:** Not applicable.

**Informed Consent Statement:** Not applicable.

**Data Availability Statement:** Not applicable.

**Acknowledgments:** The linguistic review of the present article was made by Antoanela Marta Madar, member of the Research Center "Interface Research of the Original and Translated Text. Cognitive and Communicative Dimensions of the Message", Faculty of Letters, "Dunarea de Jos" University of Galati, Romania.

**Conflicts of Interest:** The authors declare no conflict of interest.

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
