# Peer review of "Assessment of Heavy Metal Pollution Levels in Sediments and of Ecological Risk by Quality Indices, Applying a Case Study: The Lower Danube River, Romania"

_water, doi:10.3390/w13131801_

Round 1
Reviewer 1 Report
Dear Authors,
thanks for your contribution firstly. The paper provides to develop different tools for quantifying heavy metal pollution in sediments in the Danube basin and in the small adjacent rivers streams.
- I recommend reviewing the abstract, some information is redundant such as on the case study and also the purpose is not clearly defined: "a method meant to prevent this problem has been developed and different tools for quantifying heavy metal pollution have been taken into account so as to meet the needs of heavy metal management structures ", in the introduction it is explained that the purpose was to:" different specific pollution indices important in assessing the quality of big stream water bodies were used and tested ". It probably needs to be written more clearly and in accordance with the rest of the paper.
- the abbreviations of the parameters should be indicated in italics in the text;
- in line 128, paragraph 2.3.3, the abbreviation "CF" must be indicated already in that line which is the first time you mention the parameter, the same applies to the others.
- In the introduction, in addition to the purpose, it would also be useful to highlight the scientific originality of the proposed approach compared to what has already been done.
- check the manuscript there are some typos (for example the dot is missing in the abbreviation eq. 3 etc.)
- in figure 2, I suggest to remove the north indicator since already in the axes it is indicated together with east (N ordinates, and E abscissa); the graduated scale is probably not essential or if it is, it should be reduced, 100 km are too many for the image to which it refers. In the same figure, the images referring to Ni and Cd have a smaller reference scale (50 km), I recommend choosing the same scale for all images. Regarding the color graduated scale, the unit of measurement is missing.
- Finally, it might be interesting to insert a paragraph that talks about the efforts made by research to improve water quality, not only strictly related to your study type but in general. To this end, I suggest the inclusion of the following studies that deal with the optimal positioning of water quality sensors to monitor and / or prevent contamination and illegal sewer spills. In particular, the second paper, which deals with conservative contaminants (heavy metals), specifies that the quality of the wastewater has effects both on the proper functioning of the sewer system and WWTP but consequently also the receiving water bodies, especially in the case of CSOs (combined sewer overflows). doi: 10.3390 / s20123432; https: // doi.org/10.3390/w13070934.
In totally, I believe that the paper includes solid content, but the presentation need to be improved, with a better this manuscript can have its own value and impact. I hope that these recommendations are helpful to the authors and wish good luck for the further reviewing process.
Author Response
Dear Reviewer,
Thank you for the time put into revising our manuscript. We appreciate the careful review and constructive suggestions, which helped to improve our manuscript. The revision has been developed in consultation with all coauthors, and each author has approved the final form of this revision.
A detailed item-by-item/ block response (we highlighted in yellow where the text was modified in the manuscript) to suggestions follows. We hope that our answers are satisfactory and in line with your comments!
Comments and Suggestions:
- I recommend reviewing the abstract, some information is redundant such as on the case study and also the purpose is not clearly defined: "a methodmeant to prevent this problem has been developed and different tools for quantifying heavy metal pollution have been taken into account so as to meet the needs of heavy metal management structures ", in the introduction it is explained that the purpose was to:" different specific pollution indices important in assessing the quality of big stream water bodies were used and tested ". It probably needs to be written more clearly and in accordance with the rest of the paper.
Response 1: Thank you for your suggestion! We made the necessary changes [Lines 15-17].
- - the abbreviations of the parameters should be indicated in italics in the text;
Response 2: Thank you for your suggestion! We made the necessary changes.
- in line 128, paragraph 2.3.3, the abbreviation "CF" must be indicated already in that line which is the first time you mention the parameter, the same applies to the others.
Response 3: Thank you for this remark! We made the necessary changes.
4.- In the introduction, in addition to the purpose, it would also be useful to highlight the scientific originality of the proposed approach compared to what has already been done.
Response 4: Thank you for your suggestion! We have added more information to highlight the scientific originality [Lines 86-91].
- - check the manuscript there are some typos (for example the dot is missing in the abbreviation eq. 3 etc.)
Response 5: We checked the paper once more and made the necessary modifications according to the WATER requirements. Thank you for this remark!
- - in figure 2, I suggest to remove the north indicator since already in the axes it is indicated together with east (N ordinates, and E abscissa); the graduated scale is probably not essential or if it is, it should be reduced, 100 km are too many for the image to which it refers. In the same figure, the images referring to Ni and Cd have a smaller reference scale (50 km), I recommend choosing the same scale for all images. Regarding the color graduated scale, the unit of measurement is missing.
Response 6: Thank you for your suggestion! We made the recommended changes. Regarding the missing units of measurement, we specify that the calculated indices are dimensionless.
- - Finally, it might be interesting to insert a paragraph that talks about the efforts made by research to improve water quality, not only strictly related to your study type but in general. To this end, I suggest the inclusion of the following studies that deal with the optimal positioning of water quality sensors to monitor and / or prevent contamination and illegal sewer spills. In particular, the second paper, which deals with conservative contaminants (heavy metals), specifies that the quality of the wastewater has effects both on the proper functioning of the sewer system and WWTP but consequently also the receiving water bodies, especially in the case of CSOs (combined sewer overflows). doi: 10.3390 / s20123432; https: // doi.org/10.3390/w13070934.
Response 7: Thank you for your suggestion! We added information about the importance of monitoring the aquatic ecosystem quality using various analysis techniques to perform a complex environmental assessment. [Lines 58-61].
Reviewer 2 Report
In my opinion the manuscript entitled “Assessment of heavy metal pollution levels in sediments and of ecological risk by quality indices applying. A case study: the Lower Danube River, Romania” presents a comprehensive and detail report on the heavy metal pollutants in Danybe River. The manuscript is well written and the topic is within the scope of the Water Journal. Therefore It can be accepted. However I have some suggestions, which were provided below:
Lines 34-39: In my opinion, there is too little information on the adverse effects of heavy metals on ecosystems, including the harmful impact on human health. There is a lot of information available in the literature. I suggest the authors to significantly enrich this paragraph with interesting examples and data.
Table 5, Table 6 and Table 8: I suggest using the statistical analysis to assess the significance of the differences between the data corresponding to autumn and spring. Are there any suggestions, literature data/reports or analyzes suggesting what pollutant values should be expected in summer and winter?
Lines 92-98: There is a lack of information regarding analytical method accuracy and validation and other statistical details like repeatability (standard deviation, confidence intervals), the number of replicants, calibration curve details quality (including R2), recovery.
I suggest to analyze the Danube pollution reports in neighboring countries and evaluate whether this has an impact on the Romanian part of the river.
Author Response
Dear Reviewer,
Thank you for the time put into revising our manuscript. We appreciate the careful review and constructive suggestions, which helped to improve our manuscript. The revision has been developed in consultation with all coauthors, and each author has approved the final form of this revision.
A detailed item-by-item/ block response (we highlighted in yellow where the text was modified in the manuscript) to suggestions follows. We hope that our answers are satisfactory and in line with your comments!
Comments and Suggestions:
- Lines 34-39: In my opinion, there is too little information on the adverse effects of heavy metals on ecosystems, including the harmful impact on human health. There is a lot of information available in the literature. I suggest the authors to significantly enrich this paragraph with interesting examples and data.
Response 1: Thank you for your suggestion! We have added more information about the adverse effects of each heavy metal, especially on human health [Lines 40-52].
- Table 5, Table 6 and Table 8: I suggest using the statistical analysis to assess the significance of the differences between the data corresponding to autumn and spring. Are there any suggestions, literature data/reports or analyzes suggesting what pollutant values should be expected in summer and winter?
Response 2: Thank you for your suggestion! We added PCA statistical analysis (Figure 4) to assess the significance differences in heavy metal concentrations between seasons [Lines 300-307].
On the monitored sector in this paper, no studies were performed regarding the seasonal variation of the heavy metal concentrations in the Danube sediments. In a study accomplished in Serbia on Danube sediments taken in the spring, summer and autumn seasons, higher concentrations of some elements (Cu, Ni, Zn) were observed in the summer season (DOI 10.1007/s11368-015-1211-6). However, these results does not necessarily apply to the monitored sector in this paper, as the presence of heavy metal concentrations in the sediment is influenced by a number of complex factors (different sources of pollution, different flows, different hydromorphology, etc.) which differs from one area to another. In general, the winter season was not monitored in studies due to weather conditions.
- 3. Lines 92-98:There is a lack of information regarding analytical method accuracy and validation and other statistical details like repeatability (standard deviation, confidence intervals), the number of replicants, calibration curve details quality (including R2), recovery.
Response 3: Thank you for your suggestion! We have added information regarding analytical method accuracy, validation and other statistical detail (Table 1, Table 6) and [Lines 120-129].
- I suggest to analyze the Danube pollution reports in neighboring countries and evaluate whether this has an impact on the Romanian part of the river.
Response 4: Thank you for your suggestion! In order to establish whether the quality of sediments on the Romanian part of the Danube is influenced by neighboring countries, it is necessary to carry out a study on sediment transport along the Danube River. This topic is very complex, which can be developed in a separate study.
Round 2
Reviewer 1 Report
Dear Authors,
the suggestion changes have been made. Now, in my opinion, the paper is ready for pubblication
This manuscript is a resubmission of an earlier submission. The following is a list of the peer review reports and author responses from that submission.
Round 1
Reviewer 1 Report
I think the work is interesting and perfectly publishable but it would need some improvements, from my point of view, that would enrich it more.
The most important lack and problem that I see in the work is that 15 samples seem to me few to make an adequate statistical analysis. I see it as an insufficient number and very few samples. Also over 120 kms, one sample every 8 kms approximately?
The samples taken from the sediment in the 10 cm surface are from the river bank or from the river bottom?
In material and methods I would like it to be detailed how the heavy metals were extracted in the sediment, was an acid digestion carried out ? was aqua regia or another acid mixture used? Microwaves were used to digest, at what temperature? The conditions should be detailed.
Climatic data on precipitation and temperature are missing in these two sampling stations, autumn 2018 and spring 2019, precipitation can influence the washing and movements of these pollutants.
Review the missing capital letters, for example in line 196 Pollution Load Index, in the title line 195 Pollution Load Index. On line 66 Contamination Factor.
I think the spatial representations and distributions are very clear and intuitive. What do Negative IGeo values mean?
The indexes used are very suitable and interesting but they have a very important value in them, the value of the background used, it will condition all the values obtained from the indexes, depending on whether some backgrounds are used or others, the indexes will vary greatly.
In line 50 the background value for Cr is 30 and not 50 in the work of Ilie et al (2007). Check it out.
The discussion is too poor, it is more like a description of its results, something more is missing.
Part of the conclusions I think are results.

Reviewer 2 Report
The paper presented a case study of the heavy metal concentration in the surface sediment of the lower Danube river, in Romania. The samples were collected monthly for two seasons (autumn and spring) and analyzed using XRF for Cu, Pb, Zn, Cr, Ni concentration. Four known indices, Igeo, CF, PLI and RI, were used to evaluate and quantify the influence of the anthropic activities on the level of heavy metal contamination in surface sediments of the river. The manuscript revealed a local issue regarding the heavy metals in the river sediment and showed their spatial and temporal variation; however, there are a number of quite similar researches on the same issue in other nations even on the same river (references 17 and 18), this paper should contribute novel or advanced perspective on this issue to attract global readers.
Only two seasons monthly sampling data for evaluating and quantifying the anthropogenic activities’ influences are doubtful, besides, the climate, rainfall, river volume flow rate, land-use type surrounding the river, known point pollution sources, etc. also have synergetic influences on the heavy metals concentration in the surface sediments of river, which must be addressed in the manuscript.
The XRF method is generally used as a rapid monitoring method or in a pre-screen stage, of which data exist considerable variance with the data from the standard method combining sample digestion and Atomic Absorption (AA) or Inductively Coupled Plasma (ICP) spectrometer for heavy metals measurement in solid sample. The analyzed data using XRF should be calibrated with the data from standard method.
Reviewer 3 Report
1) What are the 23 heavy metals and what are its environmental significance. Provide more details and a nice table with all the recent references.
2) According to ref 9, what are these pollutants?
3) Write the novelty of this work in 5 sentences.
4) What are the previous literatures in this field and what are their limitations. Write 15 sentences about that.
5) Stat analysis of the data is missing - why? Can this work be repeated? Provide all the numbers with plus or minus values.
6) What is the effect of high Cr conc in the rivers. Provide more scientific discussion.
7) Concerning lithogenic effects, natural fluctuations of metals and some small anthropogenic influences, provide more scientific reference and facilitate ONE PAGE of discussion. Please dont only present the results. We need real scientific discussion from literatures.
8) Remove the border type box from all the figures.
9) Format all the references manually - there are TOO many mistakes and inconsistencies. Why?
10) How can Potential ecological risk index be used to formulate new policies in the region?
11) What are the existing national level river water quality policies? Compare them with international standards.
12) Write the practical applications of this work in one page.
13) Conclusions = should be less than 150 words, in one paragraph.